# Taipan Natriuretic Peptides Are Potent and Selective Agonists for the Natriuretic Peptide Receptor A

**DOI:** 10.3390/molecules28073063

**Published:** 2023-03-29

**Authors:** Simone Vink, Kalyana Bharati Akondi, Jean Jin, Kim Poth, Allan M. Torres, Philip W. Kuchel, Sandra L. Burke, Geoffrey A. Head, Paul F. Alewood

**Affiliations:** 1Institute for Molecular Bioscience, The University of Queensland, St Lucia 4072, Australia; 2Department of Pathology and Immunology, Faculty of Medicine, University of Geneva, CH-1211 Geneva, Switzerland; 3Nanoscale Organisation and Dynamics Group, Western Sydney University, Penrith 2759, Australia; 4School of Life and Environmental Sciences, University of Sydney, Sydney 2006, Australia; 5Baker Heart and Diabetes Institute, Melbourne 3004, Australia

**Keywords:** taipan natriuretic peptide, peptide synthesis, NMR structure, peptide stability, TNP pharmacology

## Abstract

Cardiovascular ailments are a major cause of mortality where over 1.3 billion people suffer from hypertension leading to heart-disease related deaths. Snake venoms possess a broad repertoire of natriuretic peptides with therapeutic potential for treating hypertension, congestive heart failure, and related cardiovascular disease. We now describe several taipan (*Oxyuranus microlepidotus*) natriuretic peptides TNPa-e which stimulated cGMP production through the natriuretic peptide receptor A (NPR-A) with higher potencies for the rat NPR-A (rNPR-A) over human NPR-A (hNPR-A). TNPc and TNPd were the most potent, demonstrating 100- and 560-fold selectivity for rNPR-A over hNPR-A. In vivo studies found that TNPc decreased diastolic and systolic blood pressure (BP) and increased heart rate (HR) in conscious normotensive rabbits, to a level that was similar to that of human atrial natriuretic peptide (hANP). TNPc also enhanced the bradycardia due to cardiac afferent stimulation (Bezold–Jarisch reflex). This indicated that TNPc possesses the ability to lower blood pressure and facilitate cardiac vagal afferent reflexes but unlike hANP does not produce tachycardia. The 3-dimensional structure of TNPc was well defined within the pharmacophoric disulfide ring, displaying two turn-like regions (RMSD = 1.15 Å). Further, its much greater biological stability together with its selectivity and potency will enhance its usefulness as a biological tool.

## 1. Introduction

Cardiovascular ailments are a major public health concern which significantly increase the risk of developing heart, brain, kidney, and other diseases [1,2]. Despite the several medical interventions in place, the clear unmet need in the effective management of cardio-renal diseases highlights the requirement for new therapeutic options to improve early diagnosis and patient care.

Natriuretic peptides (NPs) are peptide hormones that play a crucial role in the regulation and homeostasis of the cardiovascular and renal systems [3,4] through their ability to modulate vascular muscle tone, neurohormonal activity, cardiovascular remodeling, natriuresis and diuresis, and maintain intravascular volume, blood pressure, electrolyte balance, and cardiovascular homeostasis [1,2,3,4]. Their physiological significance highlights their therapeutic potential for the treatment of several cardio-renal diseases such as hypertension, congestive heart failure, and kidney dysfunction [3]. NPs also serve as effective biomarkers as well as diagnostic and prognostic tools for a variety of cardiac and renal ailments in routine clinical settings [5,6,7]. In recent years, other novel physiological functions of NPs in the metabolic system have emerged [8] that expand their therapeutic potential to include treatment of metabolic syndrome-related conditions, such as obesity and insulin resistance.

In humans, the NP system consists primarily of three main peptides, A-type or atrial natriuretic peptide (ANP), B-type or brain natriuretic peptide (BNP), and C-type natriuretic peptide (CNP). Distinct variations in the expression levels of each NP correlates strongly with normal physiological and disease conditions [3]. Due to their importance in cardio-renal diseases these peptides have been extensively studied [2,4,9]. Structurally, the NP family is characterized by a 17-residue ring structure with a highly conserved internal sequence and variable N- and C-terminal regions [10]. Whereas the conserved residues within the ring facilitate receptor binding, the variable residues in the C-terminal tail contribute to receptor selectivity [11]. NPs exert their biological effects by interacting with three membrane bound receptors, NPR-A, NPR-B, and NPR-C. NPR-A and NPR-B are guanylyl cyclase receptors and facilitate the conversion of GTP to the second messenger cGMP upon NP binding by initiating the subsequent downstream signaling cascade [12,13]. The primary role of NPR-C which does not possess guanylyl cyclase activity is in the clearance of circulating NPs through receptor-mediated internalization and lysosomal degradation [14]. Other studies have also suggested that NPR-C mediates the biological effects of some NPs (such as guanylin and uroguanylin) through a G-protein dependent method [14,15,16]. Both hANP and hBNP have short half-lives of 3 and 22 min, respectively, in plasma [11,17], which can be attributed to clearance through NPR-C binding, proteolytic cleavage by endopeptidases, such as neutral endopeptidase 24.11 or neprilysin (NEP), [18] and renal filtration [19].

Despite the therapeutic potential of these peptides, the only commercially available NP-based therapeutics are synthetic hANP (carperitide) and recombinant hBNP (nesiritide) [20,21]. These drugs are associated with a range of adverse effects and poor pharmacokinetic properties that have limited their use [22]. Other strategies for NP therapeutic development have focused on the design and synthesis of novel chimeric NPs [23] Cenderitide, a chimeric peptide consisting of CNP fused to the C-terminal tail of Dendroaspis natriuretic peptide (DNP) was designed in 2008 to eliminate the unwanted hypotensive properties of NPs while preserving the natriuretic and diuretic effects through the activation of NPR-A and NPR-B [23]. This molecule is currently undergoing clinical trials for heart failure treatment and can potentially prevent and/or reverse myocardial remodeling [24,25,26]. Thus, the potential of novel NPs for the treatment of cardiovascular conditions remains promising though as yet unfulfilled.

For several decades, snake venoms have been explored for peptides and proteins that target a range of ion channels and receptors [27,28,29]. Several components with beneficial effects for numerous indications have been uncovered, the most successful being captopril, an angiotensin-converting enzyme inhibitor from the venom of the Brazilian arrowhead viper [30,31]. These investigations also led to the discovery of several novel members of the NP family (Figure 1). Snake NPs have been reported to produce a strong hypotensive effect upon envenomation with the prey rapidly losing consciousness. The isolation of the highly potent DNP from the venom of the green mamba (*Dendroaspis angusticeps*) in 1992 stimulated further studies into NPs from snake venoms (Figure 1). DNP was shown to interact with NPR-A and NPR-C with potencies greater than that of hANP [32]. Subsequent studies have shown that snake venom NPs are similar to mammalian natriuretic peptides in terms of their structure and function, but they possess certain distinct characteristics which give them greater potency and increased plasma stability as compared to their human family members [28].

The use of -omics technologies in the past decade has allowed the comprehensive characterization of numerous snake venomes at a rapid rate [30,33,34]. Analysis of the proteomic data in a recent review showed distinct variations in the expression of different protein families with NPs being more predominant in vipers than elapids [3].

In the present study, we focus on five NPs from the venom of the inland taipan, *Oxyuranus microlepidotus* designated TNPa-e (Figure 1). Their sequences have been previously determined either from sequencing purified peptide from whole venoms [35] or cDNA isolation and subsequent sequencing [36]. Interestingly, TNPd was predicted from a single transcript from three related species including *Oxyuranus scutellatus* (coastal taipan), *O. microlepidotus* and *Pseudechis australis* (mulga snake) though not observed at the peptide level in the venom of the inland taipan. Prior research demonstrated that TNPa, TNPb and TNPc possess natriuretic activity in rat tissues [35] but their cGMP-stimulating activity was not investigated. This research characterizes TNPa-e activity at both human and rat NPR-A, evaluates their cardiovascular effects in rabbit models, and analyzes the structural features of TNPc.

## 2. Results

### 2.1. rNPR-A Activation

NP activity can be determined by the effectiveness of the peptide to elicit cGMP production through NPR-A. The cGMP ALPHAscreen evaluates the ability of peptides to stimulate cGMP production in cells through a non-radioactive, homogenous, proximity assay which encompasses bioluminescence resonance energy transfer. Previously TNPa-c were found to stimulate cGMP production and smooth muscle relaxation through rat NPRs [35]. In order to evaluate the potency of TNPa-e at rNPR-A, TNPa-e were evaluated using the cGMP ALPHAscreen in rat PC12 cells. These cells have previously been shown to express mainly rNPR-A, with minimal amounts of rat NPR-B (rNPR-B) and undetectable levels of rat NPR-C (rNPR-C) [37]. Thus, NP-stimulated cGMP production in these cells is predominantly through interaction with rNPR-A.

TNPa-e all stimulated cGMP production through rNPR-A. Their effects were compared to that of hANP and hBNP, with the concentration response curves presented in Figure 2. TNPc and TNPd were the most potent with EC_50_ values of 100 and 18 nM, respectively (Table 1). Both of these peptides were more active than hANP (EC_50_ 310 nM) and hBNP (EC_50_ 1400 nM) at this receptor. TNPa, TNPb, and TNPe had EC_50_ values greater than 1 μM.

### 2.2. hNPR-A Activation

The activity at the human receptor was evaluated using HEK 293 cells transiently overexpressing hNPR-A. As these cells have not been found to endogenously express hNPR-A, human NPR-B (hNPR-B), or human NPR-C (hNPR-C), the cGMP elevation in response to NPs can be attributed to activity through transfected hNPR-A.

In contrast to their activity at rNPR-A, TNPa-e did not significantly stimulate cGMP production through hNPR-A (Figure 3). At 10 μM, TNPb, TNPc, and TNPd caused ~30% of the maximum cGMP production compared with the responses to DNP, hANP, and hBNP. cGMP production in response to TNPa and TNPe was less than 20% of the maximum response. As a result of the low potencies, full concentration–response curves were unable to be produced.

### 2.3. Effect on Blood Pressure and Heart Rate

The cardiovascular effects of the peptides were investigated in rabbits. TNPa-c are abundant peptides in *O. microlepidotus* venom, whereas TNPd and TNPe are rare transcripts identified in cross species and have not yet been observed in the venom. Despite the slight potency increases of TNPd at hNPR-A, only TNPa-c were used for animal experimentation to best represent the inland taipan NPs. We also included DNP (from green mamba) for comparison.

TNPa-c were administered to conscious rabbits to characterize the cardiovascular effects in vivo (Figure 4 and Table 2). Average baseline systolic and diastolic blood pressures and HR were 87.9 ± 1.5 mmHg, 64.2 ± 1.2 mmHg, and 191 ± 5 bpm, respectively. During each experiment, 45 min infusions of 1, 2, and 4 μg/kg/min were intravenously administered. hANP produced a decrease in both systolic (average = −12.7 ± 0.9 mmHg, *p* < 0.01) and diastolic (−12.1 ± 0.8 mmHg, *p* < 0.01) blood pressures and an increase in heart rate (15.1 ± 3.2 bpm, *p* < 0.05) though this was not statistically different to the lesser response of vehicle (4.6 ± 4.2 b/min). Administration of vehicle in the same animals produced a small reduction in systolic diastolic and mean blood pressure (Table 2, Figure 4).

TNPc reduced both systolic (−13.2 ± 0.8 mmHg, *p* < 0.001) and diastolic (−12.5 ± 0.6 mmHg, *p* < 0.001) blood pressure with no effect on heart rate (Figure 4). TNPa and TNPb did not alter either blood pressure or heart rate compared to vehicle. We observed no differences between the effect of the doses for TNPa and TNPb peptides, but there was a borderline linear trend in MAP with the three infusions of TNPc (Table 2, *p* = 0.06). hANP also reduced both systolic (−12.7 ± 0.9 mmHg, *p* < 0.001) and diastolic (−12.1 ± 0.8 mmHg, *p* < 0.001) blood pressures which were similar to those showed by TNPc. The fall in diastolic and mean pressure was dose-related though there was no change in heart rate (Figure 4, Table 2). DNP reduced systolic (−13.3 ± 0.8 mmHg, *p* < 0.001) and diastolic blood pressure (−9.0 ± 0.9 mmHg, *p* < 0.001), but the latter was not different to vehicle. DNP had no effect on heart rate (Table 2). There were small but significant falls in systolic (−5.5 ± 0.9 mmHg) and diastolic (−5.9 ± 0.6 mmHg) pressure but not heart rate following vehicle administration. These were time-related falls in blood pressure which occurred over the course of the 3 h recording period (*p* < 0.01, Figure 4).

### 2.4. Effect on the Bezold–Jarisch Reflex Bradycardia

Bezold–Jarisch reflex assessments produced by serotonin injections were carried out before and during each of the peptide infusions and were characterized by an immediate bradycardia of −48 ± 4 bpm (average response of the three bolus doses, Figure 5). TNPc enhanced the Bezold–Jarisch reflex bradycardia, producing the greatest reduction in heart rate (−67 ± 11 bpm being the average response of the three bolus dose) at the lowest infusion of 1 μg/kg/min (*p* < 0.05). TNPa and TNPb, hANP and vehicle had no effect on the reflex bradycardia (Figure 5).

### 2.5. Stability Assays

The extended C-terminal tails of reptilian NPs render the peptides relatively resistant to proteolytic degradation [38]. The stability of TNPc in human plasma and in the presence of three common enzymes was investigated in comparison with that of hANP. TNPc was significantly more stable than hANP in human plasma, with 72% of TNPc remaining after 24 h, whereas hANP had a half-life of less than 3 h (Figure 6A).

Neutral endopeptidase 24.11 (NEP) is a zinc metalloproteinase that hydrolyses peptide bonds at the amino side of hydrophobic residues, thereby inactivating NPs by opening their ring structure. TNPc remained intact for 3 h when incubated with NEP (Figure 6B), whereas hANP was completely degraded by NEP within this time frame with 50% of hANP being degraded within 30 min. The preferential cleavage sites for TNPc after 24 h of NEP incubation were Lys 35-Phe 36, Cys 9-Phe 10, Pro 14-Leu 15, and Arg 27-Ile 28.

Plasmin and pepsin are serine and gastric acid proteases, respectively. Plasmin levels are increased in cardiovascular disease, making NP stability to this protease essential for therapeutic applications [39]. TNPc was found to be more stable than hANP to both pepsin and plasmin, with hANP being degraded by pepsin six times faster (Figure 6C) and by plasmin four times faster (Figure 6D) than TNPc.

### 2.6. Structural Analysis of TNPc

TNPc is a 39 residue polypeptide that incorporates a single disulfide bond connecting residues Cys 9 and Cys 25. The TNPc structure comprises three distinct regions, namely an N-terminal tail (spanning residues 1–8), a ring region (spanning residues 9–25), and a C-terminal tail (spanning residues 26–39). The NMR study of the TNPc molecule was undertaken to determine the flexibility of the structure. A total of 222 NOEs were obtained, consisting of 107 intra-residual, 84 sequential, 23 medium-range, and 8 long-range NOEs. Due to the small number of long-range NOEs, TNPc was found to be a highly flexible peptide which is consistent with previous NP structures [40,41,42,43,44,45]. Nevertheless, the three regions of the peptide independently adopted somewhat ordered structures as evidenced by their relatively low root mean square deviation (RMSD) values. While the RMSD of the whole peptide molecule from residues 1–39 was quite large at 3.89 Å, the ring region of the structure (residues 9–25) was well defined with a mean RMSD of 1.15 Å (Figure 7). Ten structures in the ensemble of 20 structures displayed a tendency to fold or form a bend along the regions that incorporate residues 8–11 and 19–21. The observation of these two turn-like structures in the ring region of TNPc correlated well with those found in porcine BNP (pBNP) [41] for residues 7–12 and 18–21. A more recent NMR structural study of natriuretic peptide TcNPa (from the Australian snake *Tropidechis carinatus*) showed the presence of a short six-residue α-helix in the middle of the ring region TNPc also contains a hydrophobic cluster in the cyclic portion of the molecule as found in platypus OvCNPa [44] consisting of Cys9, Phe12, Pro13, Leu14, Leu23, and Cys25. However, unlike that observed in OvCNPa, the side-chains of the hydrophobic residues in the ring region of TNPc were not all directed inwards toward the center of the ring or away from the aqueous solvent. The N- and C-terminal regions of TNPc with RMSD values of 1.43 Å and 2.25 Å respectively were less defined than the ring structure but were nonetheless more defined than the whole TNPc molecule.

## 3. Discussion

Reptilian venoms have proven to be a rich source of novel NPs that have therapeutic potential due to improved pharmacokinetic properties when compared to mammalian NPs. Previously, TNPa-e have been identified from *O. microlepidotus* venom, with TNPc being found to have similar effects as hANP on rat NPRs [35]. Here we have shown that TNPa-e differentially activate human and rat NPR-A despite sharing greater than 65% sequence identity within the intramolecular ring with the human NPs. This suggests that there are only a limited number of residues that govern NP species potency and selectivity. All five TNPa-e peptides had greater activity at rNPR-A compared to that at the human receptor, despite these receptors sharing approximately 85% amino acid identity in the extracellular domain [46]. This was particularly evident for TNPc and TNPd, with >100- and >1000-fold differences in potency respectively.

One of the main factors hindering the development of effective NP-based drugs is the paucity of a well-defined structure that could be applied in molecular modelling and SAR studies, leading to effective drug design. Early NMR studies on unbound NPs mainly reported that they adopted random coil structures, with some reports of local turn-like structures [40,41,44,45]. More recently, TcNPa isolated from *T. carinatus* venom was found to be the first member of the NP family to possess a significant secondary structure involving an α-helix. TcNPa has other interesting characteristics including an *O*-glycosylation site, activity at both NPR-A and NPR-B and strong resistance to proteolytic degradation in plasma [47]. Crystallographic structures of hNPR-C bound hCNP and hNPR-A bound hANP implicated the highly conserved residues within the NP internal ring as the likely pharmacophoric residues [15]. In particular, Phe 8 of hANP and Arg 14 of hANP were found to form critical interactions with the NPRs [15]. The corresponding positions on TNPc are Phe 10 which is in turn one and Arg 16 that lies in the middle of a hydrophobic cluster of TNPc. While there have been reports of a third helical-like turn in this region, it was not observed with TNPc. Another important residue, Gln 18 of hANP, was found to enter a shallow cavity on hNPR-A [15] while the corresponding residue in TNPc, Val 20, is located in turn 2. Therefore, the two well-defined turns observed in TNPc also appear to be present in hNPR-bound hANP and hCNP as well as unbound pBNP, indicating their importance for activity [15,41]. Further structure-activity information will be required to tease out the structural elements that are essential for activity at NPR-A in different species.

There are only a limited number of residues that vary between the taipan and mammalian NPs. The regions with the most variance include residues 4–6 and 10–12 within the intramolecular ring of TNPc, which also vary among the mammalian NPs (Figure 1). Ring residues 11–12 form part of turn 2 of TNPc, suggesting that variance in these positions may distort TNPc secondary structure. Previous research has shown that individual replacement of these residues with Ala or Gly does not significantly affect ANP activity [35]. TNPc and TNPd were the most active taipan NPs at the human receptor and possess the critical Arg residue at position 8 within the 17-residue ring. This residue is conserved throughout the human NP family and has previously been shown to be essential for both hNPR-A and hNPR-C binding [48,49]. In TNPa, TNPb, and TNPe, this Arg is replaced by His, which could explain their lower potency at hNPR-A when compared with rNPR-A. Other residues that have been shown to contribute to hANP activity at hNPR-A include Phe at ring position 2, Met in position 6, Asn in position 7, Ile in position 9, Leu in position 15 and the C-terminal tail [49,50,51]. As there is little conservation between the C-terminal tail of hANP and the TNPs, it is unlikely that the amino acid composition of the tail region significantly contributes to hNPR-A binding. This is also supported by the high potency of hANP, hBNP, and DNP at hNPR-A, despite significant sequence variability between their C-terminal tails. It appears that that the C-terminal tail contributes to several other pharmacological properties of snake NPs that make them valuable for therapeutic development. For example, the 14-residue C-terminal tail of DNP was shown to contribute to its improved half-life without affecting its affinity for hNPR-A [52,53]. Similarly, a frameshift mutation in the ANP gene renders fs-ANP (17-residue C-terminal tail) two-fold more resistant to proteolysis without affecting its affinity for hNPR-A [54]. In contrast, Na-NP (*Naja atra*, 18 residue C-terminal tail) elicits a 100-fold less potent vasodilation, compared with ANP suggesting lower hNPR-A affinity [55]. Likewise, PtNP-a (*Pseudonaja textilis*, 14 residue C-terminal tail) showed 1000-fold less potency compared with ANP, whereas Pa-NP-c (*Pseudechis australis*, 14 residue C-terminal tail) did not evoke cGMP production in kidney cells [36]. Notably, Pt-NP-a and Pa-NP-c inhibited angiotensin-converting enzyme [36]. These examples suggest that the sequence variation in the C-terminal tail influences the NP-like attributes of these peptides.

A stark example of the C-terminal extension significantly contributed to NP function can be seen in the study by Sridharan [56] on KNP. KNP was isolated from krait venom and consists of the highly conserved ring region and a long, 38 residue C-terminal tail with a putative alpha helix. These two segments of the peptide act as two pharmacophores inducing vasodilation and natriuresis through orthogonal pathways. Truncation of the C-terminal tail of KNP retaining only the highly conserved ring region called the K-ring (shares > 70% sequence identity with hANP), elevates intracellular cGMP levels through activation of rNPR-A and induces vasodilation with minimal or no diuretic effects in contrast to hANP’s ability to induce diuretic and vasodilatory activities. Re-attachment of the C-terminal tail restored the diuretic properties, in addition to enhancing rNPR-A binding and in vivo activity [56]. Recent mutagenesis studies further explored this phenomenon and identified residues at positions 3 and 14 of the intramolecular ring (in the absence of the C-terminal tail) as the key salient features for developing only vasoactive NPs. The presence of Gly 3 caused the greatest decline in heart rate and pulse pressure and its substitution with Asp these parameters resulting in a similar change in mean arterial pressure. The presence of Gly 3 and Gly 14 in the absence of the C-terminal tail is important for diuretic activity [10]. Thus, truncation of the C-terminal tail retaining only the 17-member K-ring in KNP exclusively induces hypotensive effects without altering renal output. This study suggested that it could be possible to delineate between the determinants of hypotensive and diuretic functions in NPs by examining the conserved ring structure and the C-terminal tail [10].

To conclusively identify NP residues critical for binding to human NPRs, scanning mutagenesis would be informative. SAR studies performed to date have only evaluated single mutations in specific regions of the NPs, determined the vasorelaxant activity of a Gly or D-amino acid scan of hANP, or evaluated pharmacodynamic properties of various chimeric NP mutants [57,58,59]. Several recent studies have also highlighted the importance of post-translation modifications in modulating physiological activities of NPs. Reeks [47] described the first glycosylated NP in the venom of *Tropidechis carinatus*, TcNPa, and a recent glycoproteomics study [60] discovered *O*-glycans on mature human and mammalian NPs with two glycosites in ANP positioned in the highly conserved ring region. These post-translational modifications on mature ANP were found to have a major impact on stability and circulation time of ANP as well as receptor activation in vitro and in vivo [60].

Species-specific NPR-A selectivity has previously been reported for other NPs, including hBNP and an NPR-A selective agonist APII (rANP with N- and C-terminal truncation), both of which are selective for the human over the rat receptor [46,61,62,63]. While hBNP has 316-fold reduced potency at rNPR-A as compared with hNPR-A, N-, and C-terminal truncations in APII induced greater variation in potency and efficacy toward hNPR-A compared to the rat receptor [46]. Another study focused on the development of an hNPR-A selective peptide also resulted in the generation of an hANP hexamutant species-specific hNPR-A agonist [61]. This peptide contained six point mutations within the hANP sequence, five of which were within the intramolecular ring, and was equipotent to wild-type hANP at hNPR-A but inactive at rNPR-A [61]. Comparison of the residues within TNPc, hBNP, and the hANP hexamutant highlights three residues that may be responsible for the species selectivity of TNPc and TNPd: Phe 12, Pro 13, and Val 20. Ring residues four and five are aromatic and hydrophobic residues, respectively in TNPc and TNPd (Phe 12 and Pro 13), whereas they are typically polar or positively charged in the peptides active at hNPR-A. Similarly, the corresponding residues in the hNPR-A selective peptides to Val 20 in TNPc and TNPd are typically acidic and polar in hNPR-A selective peptides. It is likely that the three mentioned TNP residues are responsible for lower activity at hNPR-A but further mutagenesis studies are required to confirm this.s

Reptilian NPs typically have elongated N- and C-terminal tails compared with the mammalian NPs. Prior research identified that these extensions increase the stability of the peptide, particularly the proline-rich C-terminal tail, through increased resistance to NEP degradation [38]. TNPa-e all possess extended, proline-rich N- and C-terminal tails and, in agreement with previous results for reptilian NPs, TNPc was found to be significantly more stable than hANP and hBNP when incubated with human and rat plasma, NEP, pepsin, and plasmin. The stability to NEP is particularly important due to the rapid inactivation of NPs when the 17-residue ring structure is opened; thus resistance is likely to increase the half-life in vivo [64]. This feature of TNPc would be advantageous for the future design of natriuretic drugs with an extended duration of action.

The in vivo effects of TNPc and hANP were characterized in commonly used animal models for natriuretic peptides. Both peptides caused a marked reduction in diastolic and systolic blood pressure. However, one of the major differences between TNPc and hANP was the effect on heart rate. The tachycardia produced by hANP is most likely a reflex response to the fall in blood pressure, and not a direct effect of the peptide itself. The lack of tachycardia in response to TNPc suggests that this peptide has a direct cardioinhibitory action or caused a possible change to the baroreceptor-heart rate reflex. The enhanced Bezold–Jarisch reflex bradycardia produced by TNPc treatment indicates increased parasympathetic activity in comparison to hANP. The animal studies thus indicate that TNPc is an effective hypotensive agent, which produces similar cardiovascular parameters, but lacks the level of tachycardia that is produced by hANP.

The similar activity of hANP and TNPc in in vivo tests is reflective of the activity of the peptides at the rabbit NP receptors. Rabbit and human NPR-A share >90% overall sequence identity, which is similar to that shared by the human and rat receptor. Thus, the activity of TNPc in the animal models may be explained by increased activity at rabbit NPR-A in comparison with human NPR-A. This study highlights the importance of in vitro testing at human NPRs in conjunction with in vivo testing in common animal models to determine the true activity of NPs.

## 4. Materials and Methods

### 4.1. Chemicals

Boc-l-amino acids were purchased from Merck (Darmstadt, Germany) and the Peptide Institute (Osaka, Japan). Boc-Pam resins were purchased from Peptides International (Jeffersontown, USA). *O*-(7-azabenzotriazol-1-yl)-*N,N,N′,N′*-tetramethyluronium hexafluorophosphate (HATU) was purchased from Genscript (Piscataway, USA). *O*-(benzotriazol-1-yl)-*N,N,N′,N′*-tetramethyluronium hexafluorophosphate (HBTU), NH_4_HCO_3_, pepsin and rat plasma were purchased from Sigma-Aldrich Pty. Ltd. (St. Louis, MO, USA). *N*,*N*-diisopropylethylamine, *N*,*N*-dimethylformamide, and trifluoroacetic acid (TFA) were purchased from Auspep (Melbourne, Australia). Plasmin was purchased from Fluka (Buche, Switzerland). NEP was purchased from R&D Systems (Minneapolis, MN, USA). ^2^H_2_O (99.9% and 99.99%) was purchased from Cambridge Isotope Laboratories (Andover, USA). Other reagents and solvents were of analytical reagent grade.

### 4.2. Peptide Synthesis

Five TNP peptides (TNPa-e) were manually synthesized using Boc chemistry via in situ neutralization solid phase peptide synthesis [65]. Syntheses were carried out on Boc-PAM resins. HATU was used as a coupling reagent instead of HBTU for difficult regions [65]. Oxidation of the pure reduced peptides (0.05 mM) was achieved using aqueous 0.1 M NH_4_HCO_3_ (pH 8.3) at room temperature for 24 h. Oxidation was monitored using analytical RP-HPLC and mass spectrometry. When oxidation was complete, the pH of the solution was lowered using TFA and the peptides were purified using preparative reversed-phase high-performance liquid chromatography (RP-HPLC).

### 4.3. Peptide Quantification

Peptides were quantified using RP-HPLC with an external reference standard as described previously [66]. Analyses were performed in triplicate using a Shimadzu analytical HPLC system (Shimadzu, Kyoto, Japan) with an Agilent Zorbax C18 column (0.21 × 5 cm, 3.5 μm) (Agilent Technologies, Santa Clara, CA, USA). UV-light detection was measured at a wavelength of 214 nm and 280 nm.

### 4.4. Mass Spectrometry

Mass spectra were acquired using an Applied Biosystems API2000 liquid chromatography/tandem mass spectrometry triple quadrupole mass spectrometer (Applied Biosystems, Carlsbad, CA, USA) equipped with an electrospray ionization source in positive ion mode (*m/z* 400–1800, with a declustering potential of 10–20 V and 0.1 Da steps). The molecular weight of the peptide was deduced from the multiply charged species using Analyst v1.4 with Bioanalyst extensions (Applied Biosystems).

### 4.5. HPLC Analysis

Analytical RP-HPLC was performed on a Shimadzu HPLC system using a Vydac C18 column (Grace Vydac, Hesperia, CA, USA; 0.46 × 25 cm, 5 μm). Separation was achieved using a linear gradient increasing at 1% solvent B min^−1^ with a flow rate of 1 mL min^−1^ over 35 min. Preparative RP-HPLC was performed on a Waters HPLC system (Waters, Milford, USA) using a Vydac C18 column (Grace Vydac, Hesperia, USA; 2.2 × 25 cm, 10 μm). A linear gradient over 35 min was used, increasing at 1% solvent B min^−1^ at a flow rate of 10 mL min^−1^. Solvent A was composed of 0.05% aqueous trifluoroacetate (TFA) and solvent B was composed of 90% acetonitrile:H_2_O with 0.043% TFA.

### 4.6. Cell Culture

Cells were cultured at 5% CO_2_ and 37 °C. HEK 293 cells were cultured in high glucose DMEM (Gibco, Carlsbad, CA, USA) supplemented with 10% FBS, and 1% penicillin/streptomycin. HEK 293 cells were seeded in media lacking penicillin/streptomycin 24 to 48 h pre-transfection. PC12 cells were cultured in high glucose DMEM (Gibco) supplemented with 10% FBS and 1% penicillin/streptomycin.

### 4.7. Transfection

HEK 293 cells were transfected using the lipofectamine 2000 transfection protocol (Invitrogen). Cells (95% confluence) were transfected with 45% pcDNA3.1/hNPR-A, 45% pcDNA3.1, and 10% pEGFP-NI to the optimal DNA concentration specified in the protocol. The co-transfection with pEGFP-NI plasmid allowed the determination of the transfection efficiency. This was assessed by the estimation of the number of GFP-expressing cells using an Olympus IX70 inverted fluorescence microscope. Transfected cells were used for assay purposes within 24 h.

### 4.8. cGMP ALPHAscreen

The cGMP ALPHAscreen was performed using the ALPHAscreen Protein A detection kit and the biotinylated cGMP supplement (Perkin Elmer, Waltham, MA, USA) with rabbit anti-cGMP polyclonal antibody (Biovision, Mountain View, CA, USA). Cells were detached from the flask and the resulting pellet was resuspended in stimulation buffer (HBSS supplemented with 0.1% *w*/*v* BSA and 1 mM IBMX at 37 °C). Cells were added to a white 384-well Optiplate (Perkin Elmer, Waltham, USA) to 30,000 cells per well for HEK 293 cells and 10,000 cells per well for PC12 cells. Cells were incubated with different concentrations of test peptides (concentration range of 1 × 10^−5^ to 1 × 10^−11^ M in stimulation buffer) at 37 °C for 30 min. Cells were lysed with lysis buffer (1 M HEPES, 0.3% Tween-20, 1% BSA in H_2_O, pH 7.4). Then 5 μL acceptor-bead mix was added to each well under subdued lighting conditions, followed by 5 μL of donor-bead mix. Plates were incubated for ~12 h, slowly shaking at room temperature. After incubation, the ALPHAscreen signal was measured using an EnVision plate reader (Perkin Elmer). The standard curve consisted of cGMP in a concentration range of 1 × 10^−6^ to 1 × 10^−12^ M. All data points were acquired in triplicate and were analyzed using GraphPad Prism 5.0 software (GraphPad Software, Inc., San Diego, CA, USA).

### 4.9. Animals

The rabbit, mouse, and rat sequences for ANP are identical and differ by one non critical position to that of human ANP. Rabbit ANP has been used for many years to investigate the mechanism of action of hANP and is therefore a most suitable species for in vivo experiments. The experiments were performed conscious without anesthetic, and because no instrumentation is used (requiring operations to implant catheters for example), multiple experiments in the same animal allowed for within animal comparisons [67,68,69].

Experiments were conducted in 12 naïve male New Zealand white rabbits, which were bred from stock and housed at the Baker Heart and Diabetes Institute. The weights ranged from 2.5 to 2.9 kg. Prior to and during experimentation, each animal was housed in individual rabbit cages under conditions of constant ambient temperature and humidity, and normal light/dark cycle (lights on from 06:00 to 18:00). Food and water were accessible ad libitum for the duration of the study. All procedures were performed in accordance with the Australian Code of Practice for the Care and Use of Animals for Scientific Purposes and were approved by the Animal Experimentation Committee of the Alfred Hospital/Baker Heart and Diabetes Institute (Number E/0263/2003B, approved 9 December 2003).

On the day of the experiment, the rabbit was placed in a standard single rabbit holding box (15 cm high and wide and 25 cm long) with wire top and raised wire grid floor. Under local anesthesia (Lignocaine HCl, 1% AstraZeneca, North Ryde, NSW, Australia), the central ear artery and marginal ear vein were catheterized. A one-hour settling period was allowed before commencing the experiment.

Pulsatile arterial blood pressure was measured with a Statham P23ID strain gauge pressure transducer (Statham, Hato Rey, Puerto Rico) and heart rate (HR) was measured by a rate-meter triggered from the arterial pulse. Mean arterial pressure, HR, and respiration rate were digitized at 500 Hz using an analog-to-digital data acquisition card (National Instruments 6024E, Austin, TX, USA) and averaged over two-second periods by computer using the LabVIEW programming language (National Instruments).

### 4.10. Heart Rate and Blood Pressure Measurements

Each rabbit received five experiments, each separated by a one-week recovery, during which a dose–response curve to intravenous (i.v.) administration of TNPa-c, hANP or a control experiment was performed, which consisted of administering the same volumes of saline 0.9% solution via the marginal ear vein catheter. For each experiment there was an initial 60 min acclimatization period and a 45 min period during which baseline cardiovascular parameters were obtained in duplicate. Dose–response curves were performed using a range of three increasing doses of the peptides administered using a variable rate infusion pump (Harvard Apparatus, Model 22 I/w, South Natick, MA, USA), at rates equivalent to the doses 3, 6, and 12 mL.h^−1^. The doses were administered at a rate of 1, 2, and 4 mg.kg^−1^ min^−1^, which were the same as those required to produce a 14% reduction in blood pressure in response to ANP. Saline (NaCl 0.9% *w/v*) solution was used as the vehicle (time control). Each dose ran for a 45 min recording period during which von Bezold–Jarisch reflex parameters were determined over the last 15 min. The order of experiments was randomized and only one dose–response curve was performed in the animal per experiment. The von Bezold–Jarisch reflexes were determined from acute falls in HR following intravenous (i.v.) bolus injection of 3, 10, and 30 μg.kg^−1^ serotonin separated by 5 min for recovery (15 min total time).

### 4.11. Drugs

5-Hydroxy tryptamine was purchased from Sigma. The doses of drugs are expressed in μg of the base. Drugs administered intravenously via the marginal ear vein were dissolved in saline solution.

### 4.12. Data Analysis

Two-second averages of all parameters were displayed on the computer and movement artefacts were excluded from the measurements. Data were averaged over at least 30 min for the control periods and over the last 10 min of administration of each dose.

### 4.13. Statistical Analysis

Values were expressed as mean ± standard error of the mean (SEM) or mean difference ± standard error of the difference (SED). For all parameters, a mixed model split unit (nested) ANOVA was performed on the change (delta) from control values. The between treatment sums of squares was partitioned into non-orthogonal contrasts comparing treatment with vehicle. In addition, the between doses sums of squares was partitioned into a linear trend (indicating a difference between doses) and the non-linear component. A Bonferroni adjustment of the t statistic was made to account for the multiple testing and lessen the likelihood of a Type 1 (false positive) error.

### 4.14. Stability

The stability of TNPc and hANP was investigated in human plasma, pepsin, plasmin, and NEP.

*Plasma stability:* Human blood (10–15 mL) from a healthy volunteer was drawn into a sealed tube containing EDTA. Plasma was separated by centrifugation at 14,000 rpm for 30 min and stored at −20 °C until use. The human plasma samples were incubated for 5 min at 37 °C prior to peptide addition. A total of 50 μL of peptide solution (5 mg mL^−1^) was added to 200 μL human plasma and incubated at 37 °C. Then 30 μL aliquots were removed at designated times, quenched with 30 μL of extraction buffer (50% acetonitrile (can), 50 μM NaCl, 1% TFA), and stored at −20 °C until subjected to RP-HPLC analysis.

*Pepsin stability:* About 100 μL of 500 μM peptide was incubated with 6 μg pepsin (activity 3.276 U mg^−1^) in 10 mM HCl (pH 2) at 37 °C. Then, 50 μL aliquots were removed at designated times, quenched by heating to 100 °C for 4 min, and stored at −20 °C until subjected to RP-HPLC analysis.

*NEP and plasmin stability:* About 100 μL of 500 μM peptide was incubated with 500 ng NEP (activity 400 pmol min^−1^ μg^−1^) or 10 μg plasmin (0.51 U mg^−1^, 0.51 μmol min^−1^ mg^−1^) in 100 mM Tris.HCl and 150 mM NaCl (pH 7.4) at 37 °C. Then, 50 μL aliquots were removed at designated times, quenched by heating to 100 °C for 4 min, and stored at −20 °C until subjected to RP-HPLC analysis.

### 4.15. NMR Structural Characterisation

Synthetic TNPc (1 mM) was dissolved in 600 μL 90% H_2_O, 10% ^2^H_2_O (*v*/*v*), pH 4.4. NMR spectra were recorded at 277 K, 283 K, and 313 K on a Bruker DMX 600 spectrometer (Bruker Biospin, Germany). The 2D experiments consisted of a TOCSY [70] using a MLEV-17 spin lock sequence with a mixing time of 80 ms, DQF-COSY [71], and NOESY [72] with mixing times of 250 ms. Solvent suppression was achieved using a modified WATERGATE [73] sequence. Structure calculations were performed using DYANA [74] and CNS [75]. The resulting structures were then visualized using MOLMOL [47,76].

## 5. Conclusions

As current drugs based on ANP and BNP have relatively short half-lives in plasma due to NEP and NPR-C-mediated degradation and renal filtration they are only available as an intravenous infusion, which makes treatment difficult outside of a healthcare setting. The development of NP-based drugs with extended half-lives would enable the use of alternative routes of administration, such as subcutaneous injection and even oral administration.

The therapeutic potential of reptilian venom components has long been recognized. It has only been in the last thirty years that snake NPs have been investigated as lead compounds for the treatment of congestive heart failure, and several have entered clinical trials. The present work has identified a class of natriuretic peptides, TNPa-e, that have properties useful for the development of longer lasting, selective, and potent biological tools.

## Figures and Tables

**Figure 1 molecules-28-03063-f001:**
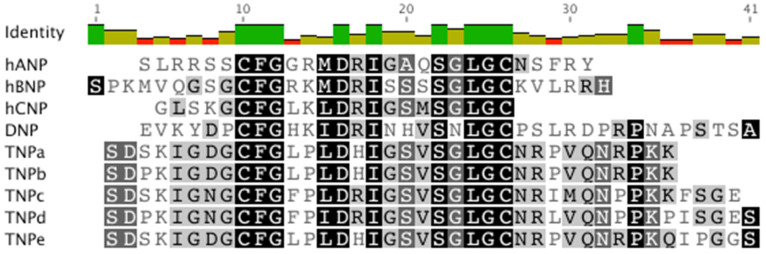
NP sequence alignment [32]. UniProtKB accession numbers are as follows (from top to bottom): P01160, P16860, P23582, P28374, P83224, P83227, P83230, Q3SAF8, Q3SAF7.

**Figure 2 molecules-28-03063-f002:**
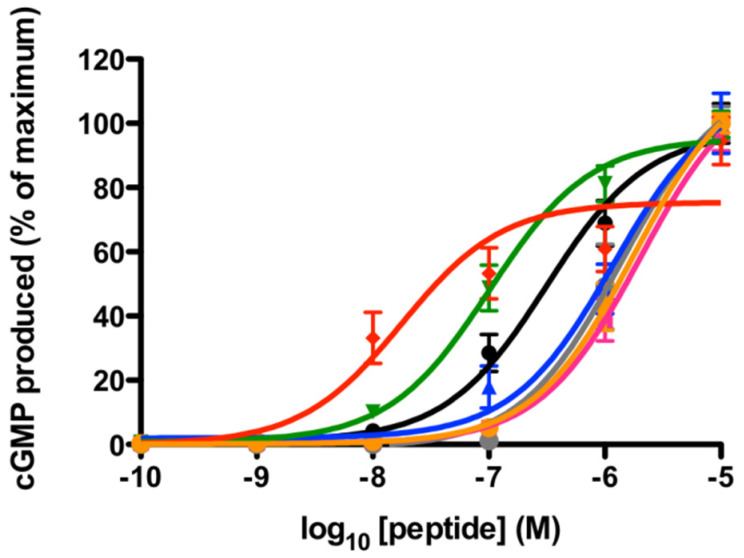
cGMP production at rat NPR-A in PC12 cells in response to TNPa-e, hANP, and hBNP. TNPd (red), TNPc (green), and hANP (black) were the most potent at rNPR-A. TNPa (pink), TNPb (blue), TNPe (orange), and hBNP (grey) had EC_50_ values of >10 μM. The data points represent the mean ± SEM of three to five separate experiments assayed in triplicate.

**Figure 3 molecules-28-03063-f003:**
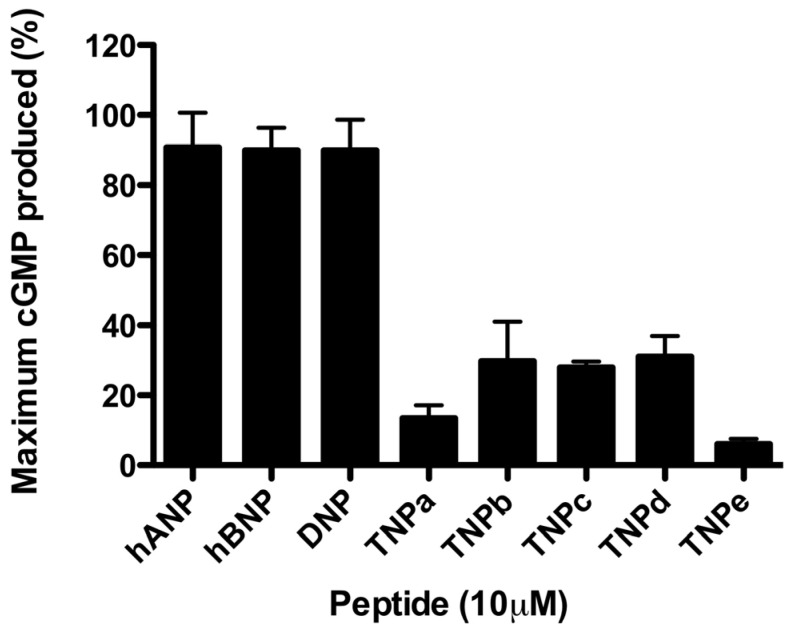
cGMP production in response to TNPa-e, hANP, hBNP, and DNP in HEK 293 cells overexpressing human NPR-A. The data points represent the mean ± SEM of three separate experiments assayed in triplicate.

**Figure 4 molecules-28-03063-f004:**
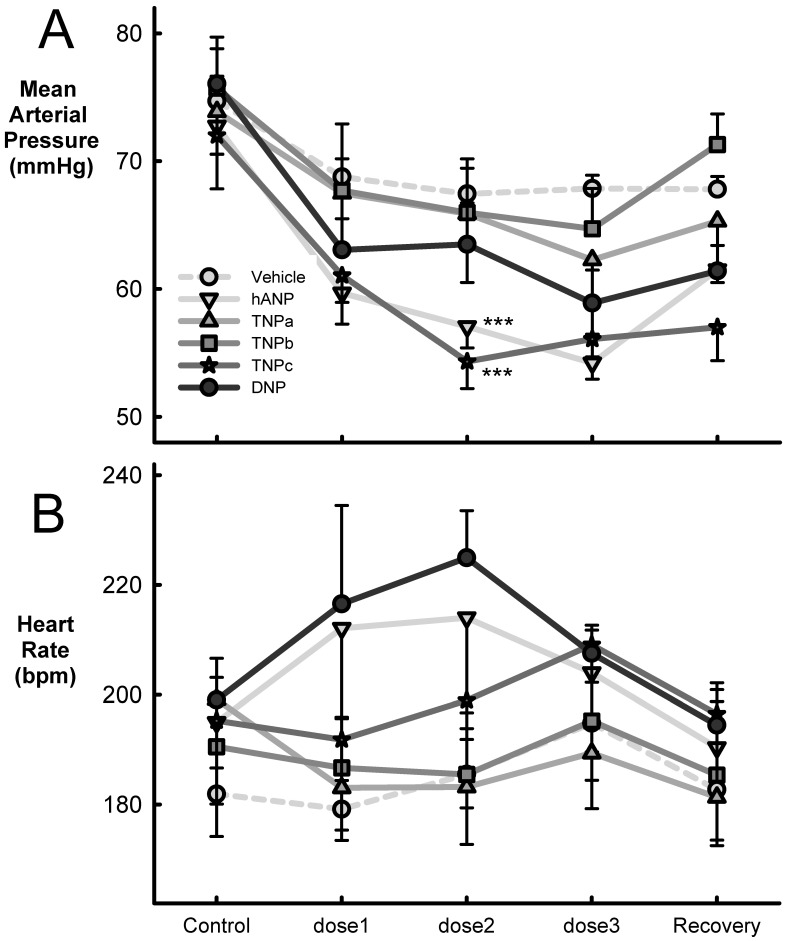
Average mean arterial blood pressure (mmHg, panel (**A**)) and heart rate (bpm, beats per minute, panel (**B**)), before (control) and after three infusions of vehicle (*n* = 10), hANP (*n* = 8), TNPa-c (*n* = 5), and DNP (*n* = 5). The doses of peptides were 1, 2, and 4 µg/kg/min for 45 min followed by a recovery period of 1 h from stopping the infusion. Error bars are SEM. *** *p* < 0.001 compared to vehicle based on the average effect of the dose.

**Figure 5 molecules-28-03063-f005:**
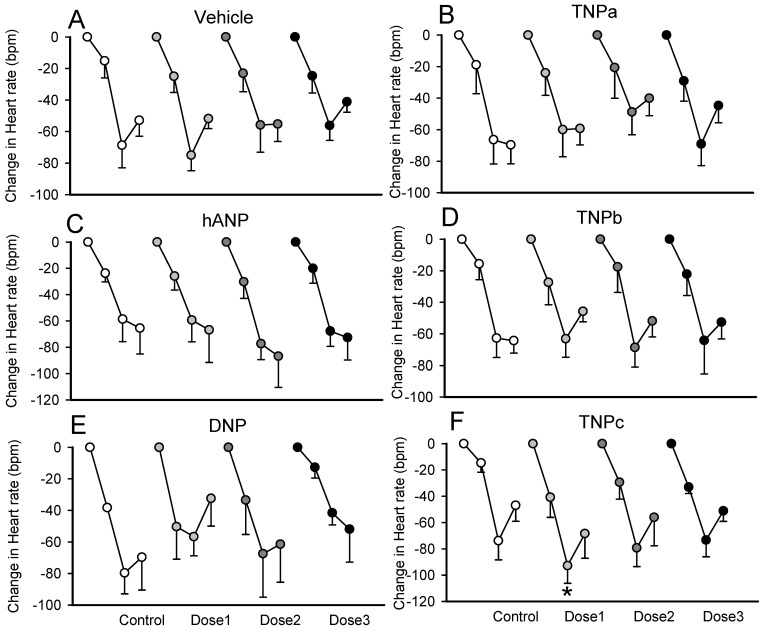
Average changes in heart rate from baseline to bolus injections of 5-hydroxytryptamine before and during infusion of three doses (dose1, dose 2, and dose 3) of TNPa,b,c (panel (**B**,**D**,**F**) respectively), hANP (panel (**C**)), DNP (panel (**E**)), or vehicle (panel (**A**)) in conscious rabbits. The change in heart rate is the response to three bolus injections of 5-hydroxytryptamine (3, 10, and 30 mg/kg i.v. data points shown for each dose). Error bars are the SE of the difference from baseline. * *p* < 0.05 for comparison of the average response during peptide infusion compared to the average response during control.

**Figure 6 molecules-28-03063-f006:**
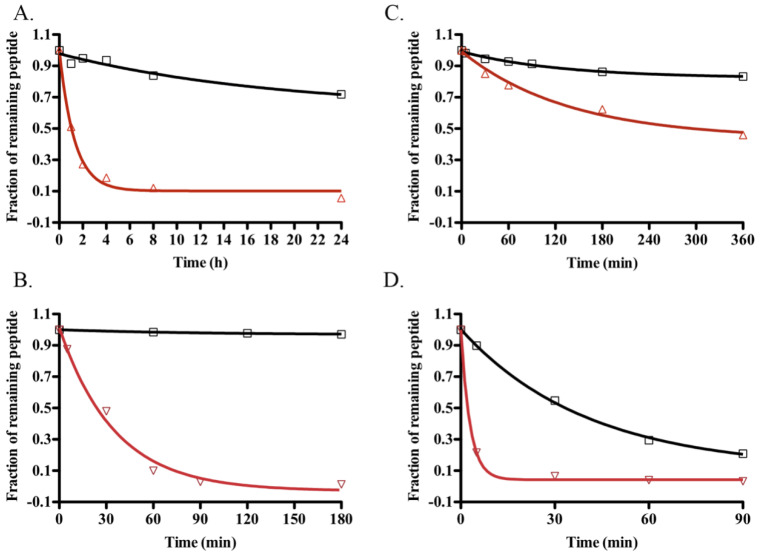
Comparative stability of TNPc (black square) and hANP (red triangle). Stability curves to human plasma (**A**), NEP (**B**), pepsin (**C**), and plasmin (**D**) were generated from RP-HPLC peak height of the remaining peptides after incubation.

**Figure 7 molecules-28-03063-f007:**
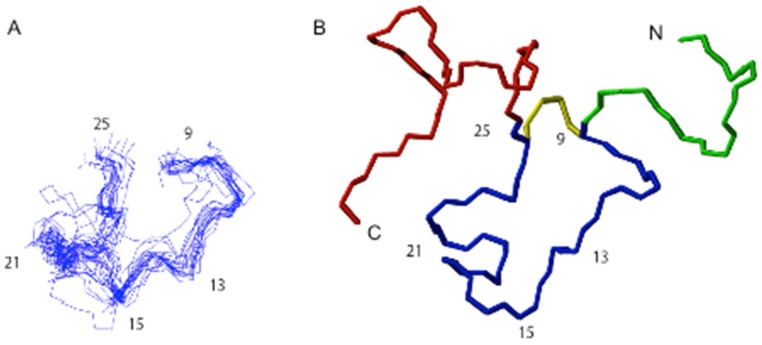
NMR structures of TNPc in aqueous solution. (**A**) Superposition of 20 lowest energy structures of the cyclic region. (**B**) Schematic diagram of TNPc tertiary structure. The distinct regions in the molecule are highlighted with different backbone colors [linear N-terminal tail (green), ring (blue) and linear C-terminal tail (red)].

**Table 1 molecules-28-03063-t001:** Potency (nM) of TNP peptides to stimulate cGMP production at rat NPR-A in PC12 cells and at human NPR-A in HEK 293 cells overexpressing hNPR-A.

Peptide	rNPR-A EC_50_ (95% CI)	hNPR-A EC_50_ (95% CI)
hANP	310 (205.1–480.2)	2.5 (1.65–3.69)
hBNP	1400 (794.6–2366)	120 (68.9–209.4)
TNPa	2020 (1453–2796)	>10,000
TNPb	1200 (729.6–1913)	>10,000
TNPc	100 (72.98–141)	>10,000
TNPd	18 (9.44–33.67)	>10,000
TNPe	1800 (1288–2439)	>10,000

**Table 2 molecules-28-03063-t002:** Cardiovascular effects of intravenous TNPa-c and DNP administration compared to vehicle and hANP in conscious rabbits.

	n	Veh1	Veh2	Veh3	SED	Between Doses	Average Response	Veh vs. Treat	hANP vs. Treat
Veh	10								
Systolic BP (mmHg)		−3.0	−5.9	−7.5	1.6	*	−5.5 ± 0.9		
Diastolic BP (mmHg)		−5.8	−6.7	−5.4	1.1	NS	−5.9 ± 0.6		
Mean BP (mmHg)		−5.9	−7.3	−6.8	1.1	NS	−6.7 ± 0.7		
HR (bpm)		−2.8	3.6	12.9	7.3	NS	4.6 ± 4.2		
		Dose (mg/kg/min)							
hANP	8	1	2	4					
Systolic BP (mmHg)		−10.6	−13.0	−14.5	1.6	NS	−12.7 ± 0.9	***	
Diastolic BP (mmHg)		−9.7	−12.3	−14.4	1.4	*	−12.1 ± 0.8	***	
Mean BP (mmHg)		−13.1	−15.7	−18.5	1.3	**	−15.8 ± 0.7	***	
HR (bpm)		17.1	19.0	9.0	5.5	NS	15.1 ± 3.2	NS	
TNPa	5								
Systolic BP (mmHg)		−2.8	−4.9	−9.4	2.3	*	−5.7 ± 1.3	NS	***
Diastolic BP (mmHg)		−6.0	−7.7	−9.9	1.6	NS	−7.9 ± 0.9	NS	*
Mean BP (mmHg)		−6.4	−8.0	−11.6	1.8	NS	−8.7 ± 1.1	NS	***
HR (bpm)		−16.1	−15.9	−9.8	6.6	NS	−13.9 ± 3.8	*	***
TNPb	5								
Systolic BP (mmHg)		−7.9	−8.8	−10.0	2.4	NS	−8.9 ± 1.4	NS	NS
Diastolic BP (mmHg)		−7.0	−7.9	−8.6	1.6	NS	−7.8 ± 0.9	NS	*
Mean BP (mmHg)		−8.0	−9.8	−11.0	1.8	NS	−9.6 ± 1.1	NS	***
HR (bpm)		−3.9	−5.1	4.7	6.0	NS	−1.4 ± 3.5	NS	NS
TNPc	5								
Systolic BP (mmHg)		−9.1	−15.6	−14.9	1.3	NS	−13.2 ± 0.8	***	NS
Diastolic BP (mmHg)		−10.0	−14.7	−12.8	1.1	NS	−12.5 ± 0.6	***	NS
Mean BP (mmHg)		−11.0	−17.7	−15.9	1.2	NS	−14.9 ± 0.7	***	NS
HR (bpm)		−3.5	3.7	13.8	6.6	NS	4.7 ± 3.8	NS	NS
DNP	5								
Systolic BP (mmHg)		−11.7	−11.9	−16.4	1.4	NS	−13.3 ± 0.8	***	NS
Diastolic BP (mmHg)		−8.5	−6.9	−11.7	1.6	NS	−9.0 ± 0.9	NS	NS
Mean BP (mmHg)		−13.0	−12.6	−17.2	1.4	NS	−14.2 ± 0.8	***	NS
HR (BPM)		17.5	25.9	8.5	11.1	NS	17.3 ± 6.4	NS	NS

Significance: * *p* < 0.05, ** *p* < 0.01, *** *p* < 0.001, NS *p* > 0.05.

## Data Availability

Submitted to Protein Data Bank. VALIDATION ID: D_9101002146.

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
