# Peer review of "Taipan Natriuretic Peptides Are Potent and Selective Agonists for the Natriuretic Peptide Receptor A"

_molecules, 2023, doi:10.3390/molecules28073063_

Round 1

Reviewer 1 Report

The manuscript by Vink et al described in vitro and in vivo activities of natriuretic peptides (NP) from inland taipan (TNPa to e). These peptides are more selective for rat NP receptor A than for human NPR-A, with TNPc and TNPd among the most potent (EC50 values of 100 nM and 18 nM) against rat NPR-A. None of these peptides resulted in >30% maximum cGMP production at 10 micromolar in HEK cells expressing human NPR-A. In rabbits tested at 3 doses (45 min infusion), TNPa, TNPb, and TNPc reduced blood pressure compared to vehicle, although statistical significance was only achieved in TNPc-treated animals. Heart rate in rabbits treated with TNPc was increased compared to vehicle but to a smaller extent than ANP. TNPc appeared to be more stable than ANP in human plasma, NEP, pepsin, and plasmin. NMR structures of TNPc showed the ring structure to be more ordered than N- and C-termini, as expected.

Overall, this is a well-done study on an interesting topic and contributed to the understanding of NP structure-function relationships. The manuscript is well-written. I enjoyed reading it. I only have a few minor suggestions for the authors consider. Hopefully these comments can help to improve clarity of data presentation in some of the figures.

1. Fig. 1: Ser1 in hBNP, and C-terminal residues in DNP, TNPd, TNPe probably should not be boxed in black?

2. Fig. 2: The curve fitting for TNPd is not as good as the rest because of the small change in response at 1 micromolar, any idea why this peptide is behaving differently compared to others?

3. Table 2: For the ‘Veh vs Treat’ comparison, is the comparison made using the average response? Or rather my question is why are comparisons not made at each individual dose? Fig. 4 appeared to suggest maybe comparison were made at individual dose since *** were marked at data point for dose 2 but Table 2 did not reflect this – so there is some disparity between Table 2 and Fig. 4 in terms of how the statistics are presented.

4. Fig. 4: I would suggest to better convey the experimental design, perhaps x-axis for the plots here can be in continuous time scale with each dose indicated and separated by the recovery/washout period? This way both experimental timeline and effect of different dose can be clearly conveyed in the same plot, which will help readers to better understand the whole experiment. If it is not going to complicate the plot, maybe also indicate the injections of serotonins?

5. Fig. 5: This is a little difficult to understand, perhaps better labels or legends will help? Eg. I am assuming Dose1 etc = NP with each successive data points in each segment representing different doses of serotonin? Also, I am having difficulty in relating the numbers reported in the text with the plots. Eg. -48+/-4 bpm refer to which data point? The text noted -67 bpm for TNPc at dose1 but the plot seems to show around -95 bpm?

Author Response

Article:

Taipan natriuretic peptides are potent and selective agonists for the natriuretic peptide receptor A

Authors:

Paul Alewood * , Simone Vink , Kalyani Akondi , Jean Jin ,Kim Poth , Allan Torres , Philip Kuchel , Sandra Burke ,Geoffrey Head *

Response to Referees

Fig. 1: Ser1 in hBNP, and C-terminal residues in DNP, TNPd, TNPe probably should not be boxed in black?

Response: I think keeping those residues in black box is OK – it is informative ad alerts the reader to similar residues between sequences.

  1. Fig. 2: The curve fitting for TNPd is not as good as the rest because of the small change in response at 1 micromolar, any idea why this peptide is behaving differently compared to others?

Response: We don’t know why the small change.  I’ve reported the experimental data as observed.

  1. Table 2: For the ‘Veh vs Treat’ comparison, is the comparison made using the average response? Or rather my question is why are comparisons not made at each individual dose? Fig. 4 appeared to suggest maybe comparison were made at individual dose since *** were marked at data point for dose 2 but Table 2 did not reflect this – so there is some disparity between Table 2 and Fig. 4 in terms of how the statistics are presented.

Response: The statistical analysis was performed on the average of the doses and not individual doses. The reason is that individual comparisons would drastically inflate the likelihood of a type one error due to making too many tests. Our hypothesis was to compare the effect of each peptide which is best represented by the average dose. We did not power the experiment sufficiently to be able to test individual doses, that would have required far more animals to be used which seemed unnecessary. Thus we chose a simpler hypothesis. We have taken the advice of professor John Ludbrook who wrote about this issue in the section Multiple pairwise comparisons of repeated measurements in his article, “Ludbrook J. Repeated measurements and multiple comparisons in cardiovascular research. Cardiovasc Res 1994;28:303-311.” He stated “In summary, the risk of experimentwise type I error is increased in approximate proportion to the number of pairwise contrasts that are made,…..”

The *** in Fig 2 do not belong to the second dose and reflect the average dose statistic but we agree this is confusing. We have altered the legend.

  1. Fig. 4: I would suggest to better convey the experimental design, perhaps x-axis for the plots here can be in continuous time scale with each dose indicated and separated by the recovery/washout period? This way both experimental timeline and effect of different dose can be clearly conveyed in the same plot, which will help readers to better understand the whole experiment. If it is not going to complicate the plot, maybe also indicate the injections of serotonins?

Response: I regret that there was an error in the figure legend which suggested there was a recovery between each dose and that each dose was infused for 15 minutes. This was not the case actually and we apologise for the error. The method section describing the protocol was correct with each infusion being for 45 minutes and then the next infusion started. There was a one hour recovery period at the end. Thus the layout of the figure closely resembles the actual time course and I have an example from an individual rabbit included below. It would overly complicate the figure to add arrows for the serotonin injections.

We have corrected the figure legend to reflect the correct design.

  1. Fig. 5: This is a little difficult to understand, perhaps better labels or legends will help? Eg. I am assuming Dose1 etc = NP with each successive data points in each segment representing different doses of serotonin? Also, I am having difficulty in relating the numbers reported in the

text with the plots. Eg. -48+/-4 bpm refer to which data point? The text noted -67 bpm for TNPc at dose1 but the plot seems to show around -95 bpm?

Response: We have corrected the legend to make it clear about the injections of serotonin at each infusion of peptide. The numbers again refer to the average reponse of the 3 doses of serotonin. We have clarified this in the text.

Reviewer 2 Report

Inland Taipan, one of the most venomous snakes in the world, has a cocktail of peptides in its venom. The current paper, "Taipan natriuretic peptides are potent and selective agonists for the natriuretic peptide receptor A" studies the natriuretic peptides, increasing sodium excretion from kidneys, therefore decreasing the cardiovascular (CVS) load and can be used as one of the potential CVS therapeutics.

The author synthesized these 5 peptides and quantified them using reverse phase C-18 HPLC. Cardiovascular effects of intravenous TNPa-c were studied in animals (rabbits). The metabolic stability of these peptides is also studied in human plasma, pepsin, plasmin, and NEP.

Although most of the important structural features of these peptides are compared and studied in the discussion section of the manuscript but the author could use these peptide sequences further to develop structural activity relationships that would improve the readability of the paper.

For example, TNP (a, b, and e) has IGDG, whereas TNP (c, and d) has IGNG, while another example is TNP (a, b, and e) has LDHI, whereas TNP (c, and d) has LDHI.

The paper is written in a systematic way and can be considered in the journal in its current form.

Author Response

There are no points to address in the manuscript though the reviewer made useful suggestions for future research